# The Long-Distance Transport of Some Plant Hormones and Possible Involvement of Lipid-Binding and Transfer Proteins in Hormonal Transport

**DOI:** 10.3390/cells13050364

**Published:** 2024-02-20

**Authors:** Guzel Akhiyarova, Ekaterina I. Finkina, Kewei Zhang, Dmitriy Veselov, Gulnara Vafina, Tatiana V. Ovchinnikova, Guzel Kudoyarova

**Affiliations:** 1Ufa Institute of Biology, Ufa Federal Research Centre of the Russian Academy of Sciences, Prospekt Oktyabrya, 69, 450054 Ufa, Russia; akhiyarova@rambler.ru (G.A.); veselov@anrb.ru (D.V.); vafinagh@mail.ru (G.V.); 2M.M. Shemyakin & Yu.A. Ovchinnikov Institute of Bioorganic Chemistry, Russian Academy of Sciences, Miklukho-Maklaya Str. 16/10, 117997 Moscow, Russia; finkina@mail.ru (E.I.F.); ovch@ibch.ru (T.V.O.); 3Zhejiang Provincial Key Laboratory of Biotechnology on Specialty Economic Plants, College of 10 Life Sciences, Zhejiang Normal University, Jinhua 321004, China; kwzhang@zjnu.edu.cn

**Keywords:** phytohormones, abscisic acid, cytokinins, jasmonic acid, lipid-binding and transfer proteins, long-distance transport

## Abstract

Adaptation to changes in the environment depends, in part, on signaling between plant organs to integrate adaptive response at the level of the whole organism. Changes in the delivery of hormones from one organ to another through the vascular system strongly suggest that hormone transport is involved in the transmission of signals over long distances. However, there is evidence that, alternatively, systemic responses may be brought about by other kinds of signals (e.g., hydraulic or electrical) capable of inducing changes in hormone metabolism in distant organs. Long-distance transport of hormones is therefore a matter of debate. This review summarizes arguments for and against the involvement of the long-distance transport of cytokinins in signaling mineral nutrient availability from roots to the shoot. It also assesses the evidence for the role of abscisic acid (ABA) and jasmonates in long-distance signaling of water deficiency and the possibility that Lipid-Binding and Transfer Proteins (LBTPs) facilitate the long-distance transport of hormones. It is assumed that proteins of this type raise the solubility of hydrophobic substances such as ABA and jasmonates in hydrophilic spaces, thereby enabling their movement in solution throughout the plant. This review collates evidence that LBTPs bind to cytokinins, ABA, and jasmonates and that cytokinins, ABA, and LBTPs are present in xylem and phloem sap and co-localize at sites of loading into vascular tissues and at sites of unloading from the phloem. The available evidence indicates a functional interaction between LBTPs and these hormones.

## 1. Introduction

The successful adaptation of plants to a changing environment depends on the efficiency of mechanisms that coordinate the growth and development of their distal organs. Phenotypic plasticity exhibited by plants in response to environmental changes requires signaling from roots to shoots and vice versa, which mediates a systemic response at the level of the entire plant [1]. Plant roots sense changes in the soil environment and generate signals that propagate between roots and the shoot to optimize plant performance, while plant roots in turn receive and decode feedback information from the shoot [2]. Long-distance transport of various molecules, including those that perform a signaling function, occurs through the vascular system consisting of two conductive tissues: xylem and phloem [3]. Roots may use hormones, or their precursors, to provide shoots with early warning of deteriorating soil conditions in ways that increase resilience to stress [4]. As the soil dries, signals transmitted to leaves through the xylem reduce both leaf transpiration and leaf growth [5]. In response to root-derived signals, back signals from shoots to roots regulate nutrient acquisition by roots in accordance with the nutrient demand of shoots [6].

Plant hormones can perform the function of transmitting signals over long distances [3,7,8]. This assumption is based on several facts. First, plant hormones are multifunctional and are known to influence plant growth and development, as well as many other processes occurring in plants [3,9]. Secondly, they were found to be present in both the xylem and phloem sap [10], and their concentration in the xylem and phloem changes in response to external influences. For example, the concentration of cytokinins decreases with a deficiency of mineral nutrients [8], while abscisic acid (ABA) and jasmonates accumulate in response to soil drying [11,12]. However, there are arguments against the involvement of plant hormones in systemic signaling. Thus, it was shown that ABA produced in leaves can close stomata in response to some other (for example, hydraulic) signals of water deficiency [13,14], while the expression of *ipt* genes, which control the production of cytokinins, increases in leaves due to the supply of nitrates [15]. Since the function of hormones as long-distance signals is subject to debate, the purpose of this review was to summarize arguments for and against the involvement of the long-distance transport of cytokinins in signaling mineral nutrient availability from roots to the shoot. It also assesses the evidence for the role of abscisic acid (ABA) and jasmonates in long-distance signaling of water deficiency and the possibility that Lipid-Binding and Transfer Proteins (LBTPs) facilitate the long-distance transport of hormones. This function of LBTPs is supported by some literature and may be useful for better understanding hormonal signaling systems. As far as we know, none of the numerous publications on LBTPs has paid sufficient attention to this aspect of their action.

## 2. Long-Distance Signaling and Transport of Cytokinins

Cytokinins are phytohormones involved in the control of shoot and root growth, leaf formation and senescence, chloroplast development, and numerous other processes [16]. Natural cytokinins are N^6^-substituted adenine derivatives with an isoprenoid side chain. The formation of cytokinin nucleotide (isopentenyladenosine monophosphate, iAMP) from adenosine phosphate is catalyzed by isopentenyltransferases (IPT), which were initially discovered in bacteria and were observed in plants only in 2001 [17]. Hydroxylation of the side chain of iPMP converts it to zeatin nucleotide, while phosphatase activity produces cytokinin ribosides from their nucleotides, from which cytokinin bases are in turn released by appropriate enzymes [18]. Conversion of nucleotides to CK bases can also occur through a one-step reaction catalyzed by LONELY GUY (LOG) [19]. This enzyme was discovered through the analysis of rice (*Oryza sativa*) mutants that are deficient in the maintenance of shoot meristems [19].

Cytokinin signaling at the cellular level begins with binding to their specific receptors belonging to the family of sensor histidine kinases [20]. A comparison of the affinities of these receptors to various cytokinin derivatives mainly showed the highest values for cytokinin bases [21], suggesting that they are the active forms of these hormones. However, several experiments have shown a high affinity of receptors for zeatin riboside in addition to its bases [22]. The importance of ribosylated cytokinins is also supported by the discovery of their high concentration in xylem sap, which indicates their function as a transport form of cytokinins [23].

The idea that cytokinins participate in long-distance signaling was born thanks to pioneer experiments conducted by Kulaeva, who discovered cytokinin-like activity in xylem sap and its decline in plants suffering from the deficit in mineral nutrients, which accelerated leaf senescence [24]. The followers of this hypothesis identified cytokinins in xylem sap and confirmed their decline under starvation [25]. It was shown that nitrate resupply to N-deprived roots stimulates cytokinin transport to the shoot [6]. Since the expression of the gene encoding phosphoenolpyruvate was induced by nitrates supplied through the roots, whereas in detached leaves the gene was up-regulated only by cytokinins and not by nitrates ([26] and references therein), these observations suggested that cytokinins transported from roots to shoots mediate nitrate signaling [27].

The perception of nitrate signal is attributed to the carrier of nitrate NRT1.1 [28]. It has been shown that, in contrast to wild-type plants, the addition of nitrates to a nitrogen-free medium did not induce the expression of genes involved in the metabolism of nitrates in the mutant of the *NRT1.1* gene [28]. These results suggested that NRT1.1 combines the functions of a sensor and a nitrate carrier, which allows it to be called a transceptor. Cytokinin content decreased in the roots of wild-type Arabidopsis in response to nitrate starvation, whereas the roots of the mutant did not respond to the treatment [29]. The results suggested the involvement of the NRT1.1 transceptor in the control of cytokinin concentration in accordance with nitrate levels.

The identification of *IPT* genes in Arabidopsis plants enabled the construction of *GUS*-containing reporters and monitoring of gene expression in different plant tissues [15]. As a result, the expression of *IPT* genes was detected in roots, which confirmed their ability to synthesize cytokinins. However, *IPT* gene expression was also found in leaves and was further enhanced by nitrate [15]. This was contradictory to the idea that root-derived cytokinins play a major role in nitrate signaling. These results were consistent with reciprocal grafting experiments showing that increased levels of cytokinins in the root resulted in only a small increase in cytokinin levels in the xylem, while authors failed to detect any phenotypic consequences in the scion [30]. This led to the assumption of the paracrine action of cytokinins precisely at the site of their synthesis. Since the involvement of long-distance transport of cytokinins in systemic signaling has been the subject of debate, we summarize reports supporting this below.

Although isopentenyladenine (iP) derivatives can be synthesized in Arabidopsis leaves, the activity of enzymes required to convert them into zeatin derivatives is low in the shoots of these plants but high in their roots [31]. Meanwhile, the AHK3 cytokinin receptor, predominantly expressed in shoots, has a higher affinity to zeatin than iP [32]. Therefore, to acquire a higher affinity for the shoot receptor, iP derivatives must be transported to the roots, where appropriate enzymes convert them into zeatin, which is transported to the leaves. The possible circulation of cytokinins from shoots to roots is supported by the presence of this hormone in phloem sap (mainly in the iP form) [33]. Experiments with the application of exogenous iP to the wheat leaves demonstrated the loading of iP into the phloem and its delivery to the roots, where the treatment increased the concentration of cytokinins [33].

The transport of cytokinins from shoots to roots is likely to be important not only for their metabolism, but also for the control of certain processes in roots. The use of a technology that blocks symplastic connections in the phloem showed that the reduction of cytokinin levels in the phloem leads to destabilization of the root vascular pattern, thereby demonstrating a role for long-distance basipetal transport of cytokinins in maintaining a vascular pattern in the roots [34]. It was shown that nitrogen foliar feeding resulted in increased shoot-to-root transport of cytokinins through the phloem [35].

The involvement of cytokinin transport in the opposite direction (from roots to shoots) in the control of processes occurring in the leaves was demonstrated in experiments with transgenic tobacco plants over-expressing the *ipt* gene. The induction of this gene in roots increased cytokinin concentration in xylem and leaves resulting in stomatal opening and increased transpiration [36] (cytokinins are known to keep stomata in the open state [7]).

Further support for the importance of systemic cytokinin signaling was provided by several experiments [37,38,39]. It has been shown that the AtABCG14 transporter, which is expressed predominantly in roots, plays the main role in the delivery of cytokinins to the shoot. The loss of *AtABCG14* expression resulted in severe inhibition of shoot growth, which was rescued by the application of exogenous zeatin to the shoots.

The importance of acropetal (from roots to shoots) transport of cytokinins for controlling shoot growth in response to the availability of mineral nutrients was demonstrated in experiments by Landrein et al. [40]. Grafting experiments showed that systemic signaling of zeatin riboside, traveling from root to shoot through the xylem, could influence the size of the vegetative meristem. This systemic signal of zeatin riboside was further shown to mediate the adaptation of shoot apical meristem size to the availability of mineral nutrients by modulating the expression of *WUSCHEL*, a key regulator of the maintenance of stem cell niche in the shoot apical meristem [40]. This mechanism allowed shoot meristems to adapt to rapid changes in nitrate concentration, thereby modulating the rate of organ production in accordance with the availability of mineral nutrients. In addition, it was shown that AtABCG14-mediated phloem unloading through the apoplastic pathway is required for the appropriate shoot distribution of root-synthesized cytokinins in Arabidopsis [41]. AtABCG14 was suggested to participate in three steps of the circular long-distance transport of iP-type CKs: xylem loading in the root for shootward transport, phloem unloading in the shoot for shoot distribution, and phloem unloading in the root for root distribution [42].

A study of the effects of supra-optimal concentration of mineral nutrients has highlighted the importance of both local and systemic cytokinin signaling [43]. High concentrations of mineral nutrients decreased cytokinin delivery from roots to shoots, resulting in their accumulation in the roots and a decline in cytokinin concentration in the shoots of barley plants. Thus, a dual result was achieved. Since cytokinins are necessary for shoot growth [44], while their increased concentration can inhibit root growth [45], both their accumulation in the roots (local effect) and decline in cytokinin concentration in the shoots (systemic effect) can lead to inhibition of both root and shoot growth. However, the mechanism of growth inhibition by supra-optimal concentration of mineral nutrients remains elusive.

Table 1 summarizes the main aspects of cytokinin metabolism, perception, and transport. For more detailed information, readers are encouraged to read reviews on the topic [23,32].

## 3. Long-Distance Transport of Abscisic Acid

The data presented in the previous section confirmed the involvement of long-distance cytokinin transport in plant adaptation to nutrient availability. Another hormone, abscisic acid (ABA), is known to be involved in the adaptation to water deficiency. This hormone was discovered more than 50 years ago as a compound that accumulated in detached leaves when they were dehydrated [46]. Determination of its structure made it possible to carry out the chemical synthesis of the bio-active isomer of ABA and to study its effects on plants. It was discovered that this hormone induces stomatal closure [2] and influences root branching [47], root hydraulic conductivity [48], and many other processes in plants.

As suggested by initial studies and later confirmed by other studies, ABA synthesis is induced by dehydration under water deficiency [49]. It was initially thought that leaves were the only site of ABA synthesis because the production of its precursor, zeaxanthin [50], requires a lot of energy, which is more easily obtained in leaves. However, it was later shown that ABA synthesis also occurs in the roots [51]. This discovery, along with the data on increased ABA concentrations in the xylem of drought-prone plants, has led to the assumption that ABA produced in roots in response to drought may serve as a signal transmitted to the shoot [5]. It has been suggested that ABA synthesized by parts of the root system as the soil dries out (e.g., in the top layers of soil in the absence of rain) may serve as an early warning signal of impending drought, while sufficient water is still available from the wet part of the soil, and shoots do not experience its deficiency. Stomatal closure observed in partial root drying experiments showed how a signal from the root can limit transpiration, thereby allowing for water conservation [5].

This hypothesis remained rather popular for some time until it was shown that ABA can be synthesized in leaves themselves in response to hydraulic signals from drying soil [2,48]. This conclusion was based on experiments showing that drought causes higher expression of *NCED* genes encoding 9-cis-epoxycarotenoid dioxygenase, a key enzyme in ABA synthesis, in leaves (namely, their vascular tissues) of Arabidopsis than in its roots [52]. Additionally, grafting of an ABA-deficient tomato mutant onto wild-type rootstocks has been shown to restore neither stomatal conductance nor leaf ABA concentrations [53], thereby refuting the importance of root-derived ABA in maintaining its leaf concentrations. 

Although these results seemed convincing and were frequently cited, there is still some evidence in support of the involvement of root-derived ABA in long-distance signaling. Thus, it was found that an ABA transporter (AtABCG25) mediates the root-to-shoot translocation of ABA in Arabidopsis [54]. The isotope-labeled ABA tracer experiments and hormone quantification in xylem sap showed that the root-to-shoot translocation of ABA was significantly impaired in the *atabcg25* mutant. The contents of ABA in the leaves were significantly lower in the *atabcg25* mutant than in wild-type plants under non-drought conditions. Consistently, stomatal closure was suppressed in the *atabcg25* mutant.

It was also shown that the expression of the genes involved in the control of ABA metabolism was higher in roots than in shoots of barley plants grown in soil, while *NCED2* expression was up-regulated in roots of plants exposed to salt-induced water shortage for 7 days [55]. In addition, Figure 3 from Holbrook et al. [53] shows that, although grafting an ABA-deficient scion onto wild-type rootstock did not increase leaf ABA content to wild-type levels, it was still higher than that in leaves of ABA-deficient plants. The results suggest that both local and systemic ABA signaling may be important in controlling plant adaptation to water deficiency, and the relative importance of shoot- and root-derived ABA likely depends on the plant species and its growing conditions.

As with cytokinins, ABA transport through the phloem has received less attention than its transport through the xylem, although the presence of ABA in the phloem was detected and described in the context of its recirculation in plants [56]. However, several reports demonstrate the importance of ABA transport from shoots to roots for the control of plant responses [57]. Thus, studies were conducted to determine whether foliage-derived ABA had an influence on root growth. Using ABA-deficient tomato and pea mutants, researchers showed that both ABA levels in roots and root growth are controlled by ABA synthesized in the leaves. Plants with ABA-deficient scions had lower concentrations of ABA in roots and significantly reduced root biomass. In contrast, plants with wild-type scions had normal root ABA content and significantly greater root mass when they were self-grafted or grafted to ABA-deficient rootstocks [57].

Shoot-derived ABA may also be important for the control of root hydraulic conductance. This hormone is known to increase the ability of root tissues to conduct water due to an ABA-induced increase in activity water channels (aquaporins) [58]. An increase in water flow from the roots to the shoots of wheat plants was detected under air warming, which helped plants maintain their leaf hydration despite increased transpiration [59]. This effect was accompanied by a rapid increase in ABA concentration in phloem exudates and roots and an increase in root hydraulic conductance. The inhibition of phloem transport by cooling the shoot base of wheat plants prevented both the accumulation of ABA in the roots of treated plants and the increase in hydraulic conductance [59]. The results indicate that shoot-derived ABA can be the source of its accumulation in roots, leading to changes in the water-conducting capacity of roots.

Thus, there is experimental evidence that the systemic effects of ABA on stomatal and hydraulic conductivity contribute to the maintenance of water balance under a changing environment. As in the previous cases, grafting experiments have been useful in this respect.

## 4. Long-Distance Transport of Jasmonic Acid, Jasmonates, and Related Oxylipins

Jasmonic acid (JA) and its derivatives, collectively referred to as jasmonates [JAs] and related oxylipins, are formed from polyunsaturated fatty acids by their oxygenation. Long-distance transport of jasmonic acid (JA) and its derivatives was mostly studied and discussed in regard to wound-induced systemic response/resistance ([3] and references therein). It was shown that necrotrophic pathogen infection, insect herbivory, and mechanical wounding induce an accumulation of jasmonates at the site of injury as well as in distal, undamaged leaves, and this triggers protective responses in both of them [60]. The results suggested transmission of a mobile wound signal from damaged leaves to distal ones, and it seemed logical to attribute this role to jasmonates. However, a more complex mechanism has been discovered involving an electrical signal, which mediates stimulation of JA production in distal leaves themselves [61,62]. Nevertheless, the translocation of jasmonates from local to distal leaves has been found in Arabidopsis, a process regulated by jasmonate transporters expressed in the phloem and localized in the plasmalema [63]. Thus, both mechanisms are likely at work in damaged plants, including the direct transport of jasmonates over long distances and induction of their synthesis in the distal leaves by some other signals coming from damaged leaves. It has been shown that JA precursor OPDA (oxo-phytodienoic acid) and its derivatives, but not the biologically active form of JA-Ile, could be transferred from wounded shoots to healthy roots [60].

Jasmonates have been shown to be involved not only in responses to biotic but also abiotic stresses [12]. Wounding as well as light and heat stress and other local stimuli can induce electrical signals to propagate over long distances, mediating hormone levels in distant organs [62]. It was shown that electrical signals are accompanied by changes in pH, Ca^2+^, and ROS levels, which induce changes in the activity of downstream responders regulating JA biosynthesis. In several studies, JA signaling was associated with the alleviation of drought stress. Endogenous JA content increased rapidly in Arabidopsis [64] and citrus [11] under drought conditions, and treatment with methyl jasmonate (MeJA) increased drought tolerance in rice [65] and soybean [66]. The increase in drought tolerance in plants treated with JA was associated with osmotic regulation and activation of the antioxidant system induced by this hormone [12]. The application of MeJA increased the concentrations of soluble sugar, amino acids, including proline, and activated antioxidant enzymes (superoxide dismutase, peroxidases, and catalase) in maize (*Zea mays*) plants and cauliflower [67].

Enhanced drought tolerance also depends on the ability of jasmonates to control water relations by closing stomata, thereby providing a water-saving strategy [68]. This mechanism is reminiscent of what is known for ABA, and the interaction of these two hormones in the control of stomatal conductance is discussed below [69]. MeJA has been shown to enhance drought tolerance in bean and barley plants by regulating stomatal closure. Jasmonates are able to regulate not only water loss, but also its flow to the leaves by influencing the hydraulic conductance of tissues [70]. It was found that exogenous MeJA increased the hydraulic conductance of bean, tomato, and Arabidopsis roots. JA-deficient tomato roots had lower hydraulic conductance than wild-type plants, while it was restored by treatment with exogenous MeJA, accompanied by increased activity of cell membrane aquaporins, PIP2 [70].

The topic of systemic signaling arises in articles devoted to the participation of jasmonates in responses not only to the biotic stresses mentioned above, but also to abiotic stresses. As in the previous cases, the use of hormone-deficient mutants and grafting experiments were useful in this regard. It was shown that wild-type rootstocks significantly increased drought-induced jasmonate accumulation in JA-deficient scions, supporting the potential for jasmonate transport from roots to shoots under drought conditions [69].

Experiments involving treating part of the root system with PEG (a model simulating partial irrigation management used in agriculture as a water-saving method) also support the notion of long-distance transport of ABA [71]. They showed that the increase in hydraulic conductivity in the roots not treated with PEG (hydrated roots) may be associated with the transport of jasmonates from the shoot [71]. This was suggested by the elevated content of jasmonates and up-regulation of JA biosynthesis genes in the leaves of the plants compared to those of the PEG-free control. The concentration of jasmonates also increased in the hydrated roots despite an unchanged expression of JA genes in the roots. The possibility of JA transport from leaves to roots was confirmed by experiments with the application of exogenous JA, which showed an increase in both jasmonate content in roots and their hydraulic conductivity, while down-regulation of the genes responsible for JA synthesis had opposite effects. These results seem to suggest that jasmonates in hydrated roots are primarily transferred from leaves through the phloem, thereby increasing hydraulic conductivity in hydrated roots in split-root experiments.

Thus, the analysis of the literature demonstrates the importance of the long-distance transport of jasmonates for the control of responses to both abiotic and biotic stresses.

## 5. Lipid-Binding and Transfer Proteins

The role of specific hormone carriers, in particular ATP-binding cassette (ABC) transporters, in regulating long-distance hormonal transport has become a popular topic of many studies and reviews [37,39,42,52] and is mentioned in this review. But for more detailed information, we recommend reading articles dedicated to this particular issue. Here, we would like to discuss another mechanism possibly implicated in the long-distance transport of hormones, which involves Lipid-Binding and Transfer Proteins (LBTPs). It is assumed that proteins of this type provide the necessary solubility of hydrophobic substances in the hydrophilic spaces and enable their movement throughout the plants [72]. Since ABA, cytokinin bases, JA, and several other hormones are considered lipophilic [73], it is not surprising that the ability of LBTPs to bind these hormones has been found and reported in several articles [74,75,76,77]. Below we present evidence of a link between the functioning of hormones and LBTPs. First, we discuss the characteristics of LBTPs.

Plant Lipid-Binding and Transfer Proteins include several classes of proteins, of which pathogenesis-related proteins class 10 (PR-10) and 14, also known as lipid transfer proteins (LTPs), are capable of binding plant hormones [72,74]. They are characterized by the presence of a hydrophobic cavity, inside which the ligand-binding site is located [72]. These proteins are able to reversibly bind various hydrophobic molecules of different chemical structures, including plant hormones such as JA, ABA, and cytokinins. The ability of LTPs and PR-10 proteins to bind plant hormones was demonstrated using a fluorescence binding assay. In addition, the spatial structures of complexes of several PR-10 proteins with cytokinins were obtained using NMR spectroscopy and X-ray crystallography (Table 2).

Many studies have shown the participation of both hormones and LBTPs in plant adaptation to biotic and abiotic stress, as well as the existence of a relationship between their functioning. Members of the LTP and PR-10 classes are responsive to many abiotic and biotic stressors, including drought and salt stresses [88,89]. For example, under salt stress, a higher transcript accumulation of LTPs was detected in the salt-tolerant variety of durum wheat, compared to the sensitive one, while the over-expression of this gene in Arabidopsis promoted plant growth under various stress conditions and enhanced plant resistance against pathogenic fungi [90]. Rapid up-regulation of PR-10 transcription in peanut callus has been demonstrated under salinity, osmotic stress, heavy metals, and low temperatures using real-time PCR experiments [91]. Altogether, these data suggest the involvement of LBTP genes in the tolerance mechanisms to both abiotic and biotic stresses in plants. Meanwhile, it is known that an increase in plant resistance is induced by stress hormones such as ABA and JA (see the previous section of this review) [60,92].

JA and ABA have also been widely shown to be involved in the regulation of *LTP* and *PR-10* gene expression [89,93]. An increased level of LTPs was detected with the help of immunolocalization in the salt-stressed roots of pea plants, which coincided with increased deposition of suberin in Casparian bands [77] and was in accordance with the known ability of LTPs to bind suberin precursors [94]. The linkage with hormonal systems, in this case, was that ABA affected LTP levels and suberin deposition in a manner similar to the effects of salinity. 

Massive LTP secretion detected during somatic embryogenesis in grapes or carrots suggests their involvement in plant development mechanisms [95]. LTP genes were expressed in the endosperm, embryo, and/or surrounding areas during seed development and germination [96]. At the same time, it is known that embryogenesis, as well as the development and germination of seeds, depend on plant hormones [97].

Several articles have shown the important role of LBTP protein-hormone complexes in plant stress defense. LTPs encoded by the *DIR1* gene (defective in induced resistance) have been shown to be required for long-distance signaling during systemic acquired resistance (SAR), while its gene mutation results in defective induced resistance [98]. However, transgenic plants over-expressing the *DIR1* gene did not exhibit constitutive SAR, suggesting that *DIR1* is probably not a mobile signal itself, but rather acts in concert with a mobile co-signal that has not yet been identified, although the possibility that it is jasmonate has been considered [99]. The biological properties of tobacco LTP were compared to those of its complex with jasmonic acid [99]. Treatment with both LTP and jasmonic acid increased the resistance of tobacco plants toward *Phytophthora parasitica*, while this effect was absent upon treatment with LTP or jasmonic acid alone. This work demonstrates the importance of binding a hormone to LBTPs. Root elongation of wild-type Arabidopsis plants was decreased dramatically by the treatment with JA compared to transgenic lines over-expressing the wheat LTP gene. It was suggested that wheat LTP may influence JA response and signaling by forming a complex with this hormone [90].

It also appears important that some of the effects of LBTPs resemble those of plant hormones. In particular, a potato LTP was shown to regulate stomatal closure in plants infected with *Phytophthora infestans* by interacting with ABA receptors [100], thereby preventing further pathogen entry [101]. Moreover, it has been suggested that the molecular mechanisms of potato LTP participation in protection against pathogens may be realized through the increased expression of stress-responsive genes in salicylic acid signaling pathways [100].

The unusual function of LTPs isolated from tobacco and wheat tissues has been demonstrated. These LTPs were able to enhance the extension of cell walls in vitro, while a hydrophobic cavity in the LTP molecule was necessary for this activity [102]. It was hypothesized that LTP associates with hydrophobic wall compounds, causing nonhydrolytic disruption of the cell wall and subsequently, facilitating wall extension. This effect resembled the activity of expansins, and it is of interest that expansins have been shown to work in concert with the hormone auxin [103].

The works cited above suggest the joint participation of LBTPs and hormones in several processes in plants. An explanation for this association may be the involvement of LBTPs in hormonal transport. In the phloem and xylem sap of plants, various hydrophobic molecules such as hormones, as well as various lipid-binding proteins were found. It is known that in human blood, lipophilic hormones (steroid and thyroid hormones) are transported in complexes with blood proteins [86]. Thus, it was suggested that long-distance transport of plant hormones may also be mediated by LBTPs (Table 1). Data indicating LBTP-mediated transport of cytokinins, ABA, and jasmonates are discussed below.

The importance of hydrophilicity for the movement of molecules throughout the plants is an argument for cytokinin ribosides being a transport form of cytokinins since ribosides are more hydrophilic than cytokinin bases. However, bases of cytokinins predominate in phloem sap [23], while PR-10 proteins, whose natural ligand was found to be zeatin [74], were detected in the phloem using proteomics [85]. These results suggest that PR-10 proteins can act as transporters of hydrophobic cytokinin bases in phloem. Furthermore, hydrophobic cytokinin bases were found in xylem sap. The presence of this type of cytokinin in the xylem seems important since grafting between various cytokinin biosynthetic and transportation mutants revealed that root-to-shoot translocation of trans-zeatin controls leaf size [104]. Information on the presence of PR-10 in the xylem is limited. However, proteins of this class have been identified in grape genotypes tolerant to Pierce’s disease caused by the bacterium *Xylella fastidiosa* as novel proteins in the xylem that help overcome pathogen attacks [84]. Long-distance transport of cytokinins and the possible involvement of PR-10 proteins [105] in cytokinin transport are shown in Figure 1.

LTPs are capable of binding jasmonic acid [72], while jasmonic acid, as well as representatives of the LTP class, were found in the phloem sap of various plants [106,107] (Table 1 and Figure 1). As mentioned above, jasmonates [108] and DIR1 [99,109] can participate in long-distance signaling during systemic acquired resistance (SAR). Cell-to-cell movement of DIR1 through plasmodesmata has been shown to be important during long-distance SAR signaling in Arabidopsis [81]. A putative orthologue of DIR1 was also identified in the phloem sap of tomato plants [87]. However, the importance of LTPs for long-distance jasmonate transport requires further investigation.

Jasmonic acid [69] and LTPs [83,84,110] were found in xylem sap as well. For example, root-predominant LTP was found in the xylem sap of soybean [83]. A new family of small cysteine-rich proteins structurally similar to lipid transfer proteins has been discovered in tomato xylem [110]. One protein from broccoli and rapeseed xylem showed a high degree of identity to LTPs [84].

The possibility of interaction of jasmonates with LTPs is suggested by the results of their immunohistochemical localization with the corresponding antibodies on sections of pea roots (Figure 2). Figure 2 shows that both LTP and jasmonates were localized in the same cell walls of xylem vessels, which may be a prerequisite for their binding and facilitation of xylem loading of jasmonates. The abundance of LTP and jasmonates was increased under salinity in the cell walls of xylem cells. The mechanisms by which LTPs influence the diffusion of hormones may involve not only their binding, but also the induction of loosening cell walls [102]. This ability of LTPs to influence cell wall structure may be important in facilitating the apoplastic transport of hormones.

Abscisic acid is one of the hormones that has been shown to be a ligand for LTPs [72]. However, the importance of LTPs for long-distance transport of ABA has received even less attention than the importance of these proteins for the systemic signaling of jasmonates. Immunohistochemical localization of ABA and LTPs on the sections of pea roots showed that their abundance was increased under salinity in the cell walls of phloem cells [77]. This was interpreted as indicating the possible involvement of LTPs in the unloading of ABA from the phloem under salinity (Figure 1).

## 6. Conclusions

The reports summarized in this review support the importance of the long-distance transport of hormones. For example, they show that the disruption of cytokinin transport from roots to shoots reduces the concentration and function of these hormones in shoots, whereas cytokinin delivery from shoots to roots is required for the proper vascular development in roots. In turn, the reports collected in this review also confirm the importance of long-distance transport of ABA. They show that the impaired function of an ABA transporter in roots decreases ABA content in leaves, thereby increasing stomatal conductance, whereas shoot-derived ABA is important for the control of root growth and hydraulic conductance. Furthermore, experiments with osmotically stressed plants showed that an increase in hydraulic conductivity in the roots is associated with the transport of jasmonates from the shoot.

A novel aspect of this review is the suggestion that Lipid-Binding and Transfer Proteins (LBTPs) are involved in hormonal long-distance signaling. We assume that proteins of this type ensure the solubility of hydrophobic substances, such as some hormones, in hydrophilic spaces and their movement throughout the plant. This assumption is based on data that show the ability of LBTPs to bind hormones, the presence of LBTPs along with hormones in the sap of xylem and phloem, and their co-localization at the sites of hormone loading into vascular tissues and unloading from the phloem. This hypothesis needs to be examined further. However, we hope that the data presented in this review can justify efforts in this direction.

## Figures and Tables

**Figure 1 cells-13-00364-f001:**
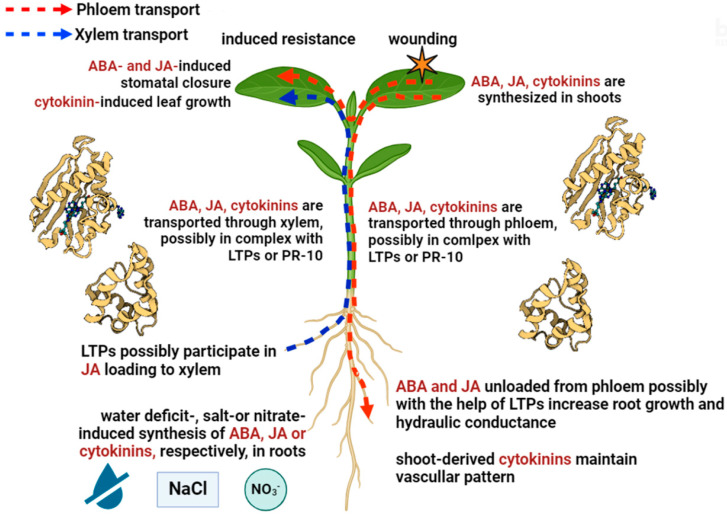
Long-distance transport and signaling of cytokinins, jasmonates, and ABA. Nitrates stimulate the synthesis of cytokinins in roots, while NaCl and deficiency of water stimulate the synthesis of jasmonates and ABA, which are transported from roots to shoots through the xylem, where cytokinins stimulate leaf growth, whereas ABA and jasmonates close the stomata. LTP possibly contributes to JA loading into the xylem and transport to the leaves. Damage to one leaf can result in the transfer of JA through the phloem to another leaf, thereby contributing to increased resistance. Transport of JA and ABA from leaves to roots up-regulates the expression of aquaporin genes, thereby increasing hydraulic conductance and influencing root growth. Nitrogen foliar feeding increases the loading of leaf cytokinins into the phloem, which possibly bind to PR-10 (PDB 2FLH, PR-10 from *Vigna radiata* forms a complex with zeatin [105]). This complex is transported from shoots to roots. PR-10 of *Vigna radiata* in complex with zeatin (PDB 2FLH) and LTP of *Pisum sativum*, capable of binding JA and ABA (PDB 2N81), are shown [72,77].

**Figure 2 cells-13-00364-f002:**
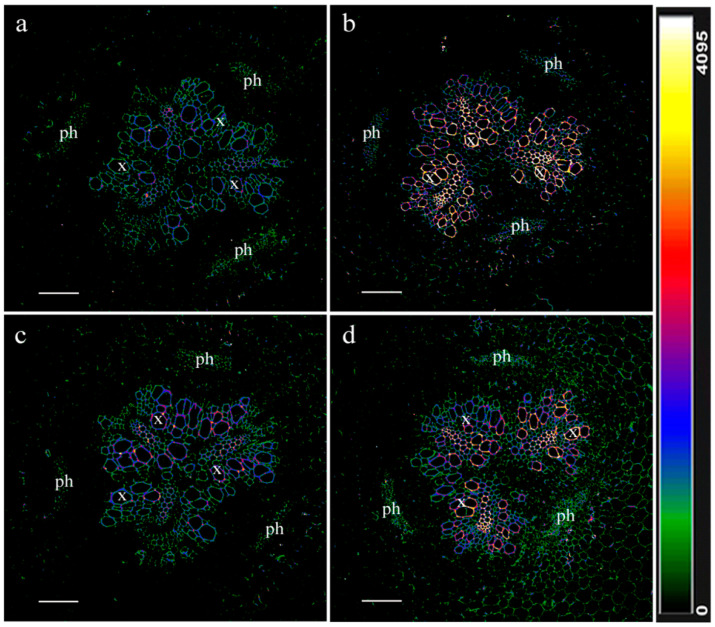
LTP (**a**,**b**) and JA (**c**,**d**) immunohistochemical localization in the central cylinder of the basal part of the roots of the control (**a**,**c**) and salt-treated (75 mM NaCl) (**b**,**d**) pea plants. The intensity of fluorescence of the second antibody against rabbit immunoglobulins is color-coded; a green color corresponds to lower fluorescence, while blue and red colors reflect a gradual increase in fluorescence. x—xylem, ph—phloem. Scale 100 µm. The principle of the experimental design is described in [77]. The novelty of our results lies in the use of antibodies against JA (Agrisera, Vannas, Sweden) to visualize the localization of jasmonates by an Olympus FV3000 Fluoview (FV31-HSD) confocal laser scanning microscope (Olympus, Tokyo, Japan) with an excitation laser wavelength of 561 nm. The results have not been published anywhere before.

**Table 1 cells-13-00364-t001:** Metabolism, transport, and perception of cytokinins.

Cytokinin Metabolism	Enzymes and Induction	Where This Process Occurs	Transport Pathways of Metabolites	Perception
Synthesis of isopentenyladenosine phosphate, iAMP, from adenosine phosphate	Catalyzed by isopentenyl transferase [17]; induced by nitrates [27]	In both shoots and roots	-	-
Dephosphorylation of iAMP resulting in the production of isopetenyladenosine (iPA)	Catalyzed by phosphotase; the reaction is reversible [18]	Both in shoots and roots	iPA is mainly transported from shoots to roots through the phloem [23]	iPA has some affinity for cytokinin receptors [22], although less than cytokinin bases [32].
Deribosylation of iPA and its conversion to isopentenyladenine (iP)	Catalyzed by adenosine nucleosidase [18]	Both in shoots and roots	iP is mainly transported from shoots to roots through the phloem [23]	iP has a higher affinity to cytokinin receptors than iPA [32]
Hydroxylation of iP and its derivative resulting in the production of zeatin	Catalyzed by cytochrome P450 mono-oxygenases [23]	Mainly in roots [31]	Zeatin riboside is the main transport form in the xylem, although zeatin is also present	AHK3 cytokinin receptor, predominantly expressed in shoots, has a higher affinity to zeatin than iP [32]
One-step conversion of cytokinin nucleotides to cytokinin bases	Catalyzed by enzyme encoded by LOG gene [19]	Both in shoots and roots	-	This reaction results in one-step production of active cytokinins with the highest affinity to their receptors

**Table 2 cells-13-00364-t002:** Some features supporting the participation of LBTPs in the long-distance transport of hormones involved in plant adaptation to the availability of mineral nutrients and water deficit.

Characteristics	LTPs	PR-10
Binding to hormones	JA [78] ABA [77]	Cytokinins [79,80] ABA [81]
Spatial structure of protein-hormone complexes	Computer modeling complex of pea Ps-LTP1 with ABA [77]	Complex of mung bean PR-10 with zeatin (PDB 2FLH) [79], complexes of PR-10 from *Medicago truncatula* with kinetin (PDB 4JHH) and trans-zeatin (PDB 4JHG) [80]
Found in xylem sap (proteomic tools)	[82,83]	[84]
Found in phloem sap (proteomic tools, immunoblot)	[85,86,87]	[85,86]
Found in xylem/phloem tissues (immunolocalization)	[77]	-

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
