# Peer review of "The Long-Distance Transport of Some Plant Hormones and Possible Involvement of Lipid-Binding and Transfer Proteins in Hormonal Transport"

_cells, 2024, doi:10.3390/cells13050364_

Round 1

Reviewer 1 Report

Comments and Suggestions for Authors

 General comments:

This review article focuses on summarizing arguments for and against the involvement of the long-distance transport of cytokinins in signaling mineral nutrient availability from roots to the shoot. It also assesses evidence for the role of abscisic acid (ABA) and jasmonates in long-distance signaling of water deficiency and the possibility that Lipid-Binding and Transfer Proteins (LBTPs) facilitate the long-distance transport of hormones. The manuscript holds scientific potential and may provide in-depth knowledge of hormones as long-distance signals for plant adaptation to the environment. However, some major points and some typographical errors need to be addressed before publication to overall enhance the quality of the manuscript. A moderate English Editing is required to improve the quality of the manuscript.

Major Comments for authors:

Title: Delete “in this Process” from the title or modify it.

L22: long-distance

L56: Delete “And finally”

Introduction: The introduction should highlight only the relevant information related to the present study.

L75” Delete . from section title.

For section 2: Add a relevant figure or a table to better understand the mechanism.

L84: What are free CK bases?

L85: Briefly describe “LONELY GUY”

L102: Delete (.

 L203: What NCED stands for?

L210: Write gene names in italic form and protein names in non-italic form. Eg. AtABCG25

Adjust the spacing errors throughout the manuscript.

Conclusion: It should highlight the major findings of the present manuscript. It should be concise and informative.

Comments on the Quality of English Language

A moderate English Editing is required to improve the quality of the manuscript.

Author Response

Comments of reviewer and our responses:

This review article focuses on summarizing arguments for and against the involvement of the long-distance transport of cytokinins in signaling mineral nutrient availability from roots to the shoot. It also assesses evidence for the role of abscisic acid (ABA) and jasmonates in long-distance signaling of water deficiency and the possibility that Lipid-Binding and Transfer Proteins (LBTPs) facilitate the long-distance transport of hormones. The manuscript holds scientific potential and may provide in-depth knowledge of hormones as long-distance signals for plant adaptation to the environment. However, some major points and some typographical errors need to be addressed before publication to overall enhance the quality of the manuscript. A moderate English Editing is required to improve the quality of the manuscript.

Response: We are most grateful to the respected reviewer for attentive reading of our article and   valuable comments. We tried to follow all of them

  1. Title: Delete “in this Process” from the title or modify it.

Response: The title was modified and “in this process” substitute with “hormonal transport”. The title was further modified according to the remark of another reviewer, who emphasized that we did not discuss all plant hormones. It now sounds in the following way: “The Long-Distance Transport  of Some Plant Hormones and Possible Involvement of Lipid-Binding and Transfer Proteins in  Hormonal Transport”

  1. L22: long-distance

Response: the dash was inserted

  1. L56: Delete “And finally”

Response: “finally” was deleted

  1. Introduction: The introduction should highlight only the relevant information related to the present study.

Response: In accordance with this recommendation we deleted two sentences in the Introduction: (1) “Early warning of water deficiency and nutrient availability thereby optimizes shoot growth and adaptive responses” (since it repeats what is said in the next two sentences) and (2) “The fact is that, unlike animals, in which production of hormones is limited to specialized glands, almost every plant cell is capable of synthesizing hormones.” We hope that this may be sufficient, since introduction is already rather short.

  1. L75” Delete . from section title.

Response: The point was deleted

  1. For section 2: Add a relevant figure or a table to better understand the mechanism.

Response: Thanks for this advice. It was interesting to try to summarize information about metabolism, transport and perception of cytokinins. It remains to hope that we managed with this. It is in Table 1 now.

  1. L84: What are free CK bases?

Response: since cytokinin bases can be present as part of their derivatives, by the term “free CK bases” we wanted to emphasize that cytokinin bases released from their derivatives are meant.  But we decided to follow remark of respected reviewer and deleted the word “free”

  1. L85: Briefly describe “LONELY GUY”

Response: According to this recommendation, after the sentence “Conversion of nucleotides to CK bases can also occur through a one-step reaction catalyzed by LONELY GUY (LOG)”, we added that “This enzyme was discovered through the analysis of rice (Oryza sativa) mutants that are deficient in the maintenance of shoot meristems [19]”

  1. L102: Delete .

Response: points at the end of section titles were deleted

  1. L203: What NCED stands for?

Response: In accordance with this remark we added that “NCED genes are encoding 9-cis-epoxycarotenoid dioxygenase, a key enzyme in ABA synthesis".

  1. L210: Write gene names in italic form and protein names in non-italic form. Eg. AtABCG25

Response: We adjusted italic form in the cases, when not transporters themselves, but genes encoding them are meant. For example, Loss of AtABCG14 expression

  1. Adjust the spacing errors throughout the manuscript.

Response: Spacing errors were adjusted

Reviewer 2 Report

Comments and Suggestions for Authors

Although several reviews on long-distance stress signalling in plants have been published in past years, I enjoyed reading this one. It summarizes the most recent knowledge about hormonal long-distance signalling. It adds an extra link to the newly emerging topic -the role of lipid binding and transfer proteins (LBTP) as modulators of signalling in plants. This is really novel part of the review and makes it unique. The first few chapters about general long-distance signalling are necessary for a complete understanding of the relationships between hormonal actions and LBTP described in the second part of the review. It is well-written with sound logic and style and is generally helpful for a broader audience.

Surprisingly, the published information on ABA and LTPs (pg.12) is limited compared to other hormones. Are there really no more relevant papers?

There are only a few minor corrections necessary (see below).

L. 153 change to "acropetal"

L 418 isolated line should be formatted better

L 423 the species name “Vigna radiata” should be in Italics

Comments on the Quality of English Language

The language is generally good both in spelling and style. Only minor corrections are needed:

 L.21 I suggest “long-distance transport.”

L 52 “It is believed…” the sentence style should be improved.

L  290 “in” can be deleted

Author Response

Comments of reviewer and our responses:

Although several reviews on long-distance stress signalling in plants have been published in past years, I enjoyed reading this one. It summarizes the most recent knowledge about hormonal long-distance signalling. It adds an extra link to the newly emerging topic -the role of lipid binding and transfer proteins (LBTP) as modulators of signalling in plants. This is really novel part of the review and makes it unique. The first few chapters about general long-distance signalling are necessary for a complete understanding of the relationships between hormonal actions and LBTP described in the second part of the review. It is well-written with sound logic and style and is generally helpful for a broader audience.

Response: It was a pleasure to read this review. Thanks!

Comment of reviewer: Surprisingly, the published information on ABA and LTPs (pg.12) is limited compared to other hormones. Are there really no more relevant papers?

Response: As far as we know interaction of ABA with LTP in vitro and their co-localization in vivo have been studied only by us (Akhiyarova et al., 2021; Melnikova et al., 2023 – see the list of references). We hope that our publication will draw attention to interaction of ABA and LTP

There are only a few minor corrections necessary (see below).

  1. 153 change to "acropetal"

Response: We are sorry for this mistake. It was corrected

  1. L 418 isolated line should be formatted better

Response: Format will be changed, since we are replacing 3 figures with one based on recommendation of another reviewer. We shall try to make it better.

  1. L 423 the species name “Vigna radiata” should be in Italics

Response: This mistake was corrected

Comments on the Quality of English Language

The language is generally good both in spelling and style. Only minor corrections are needed:

  1. 21 I suggest “long-distance transport.”

Response: The sentence was changed. It is now as follows: “Long-distance transport of hormones is therefore a matter of debate”

  1. L 52 “It is believed…” the sentence style should be improved.

Response: The sentence was modified: “Plant hormones can perform the function of transmitting signals over long distances”

  1. L  290 “in” can be deleted

Response: “In” was deleted

Reviewer 3 Report

Comments and Suggestions for Authors

The manuscript by Akhiyarova et al. reviewed the literatures on long-distance transport of plant hormones and the possible involvement of lipid-binding and transfer proteins in this process. In general, the review was well written. However, I recommend to combine figure 1, 3 and 4, which are very similar in general. Furthermore, evidence to support a role of lipid-binding and transfer proteins in long-distance transport of these phytohormones is scarce, please combine 5.1-5.3 and clarify that the involvement of LBTPs of long-distance transport of these plant hormones is speculative.  The manuscript concerns the long-distance transport of plant hormones most under abiotic averments but not biotic environments, please clarify.      

 .

My recommendations for revisions

1.     The tittle of the manuscript is grammatically incorrect and can be changed to “The long-distance transport of some plant hormones and the possible involvement of lipid-binding and transfer proteins in this process”, given that Auxins, SAs and GAs were not considered by the manuscript.

2.     L 75, the subtitle should be “Long-distance signaling and transport of cytokinins ”.

3.     L251, in this section, jasmonates (JAs, jasmonate and related oxylipins) should be used and information on the shoot-root transport of JA precursor OPDA should be included.   

4.     L109, the involvement of JA in SAR is questionable, the reference 109 do not support such a claim.

         5.   L 440-443, the conclusion is not justified by the evidence.       

Comments on the Quality of English Language

The Engish is gernerally well ecept for some  grammar misusages and misspellings. 

Author Response

We are most grateful to the respected reviewer for attentive analysis of our article and valuable comments. We have made of best to follow them.

Comments of reviewer and our responses:

The manuscript by Akhiyarova et al. reviewed the literatures on long-distance transport of plant hormones and the possible involvement of lipid-binding and transfer proteins in this process. In general, the review was well written.

  1. However, I recommend to combine figure 1, 3 and 4, which are very similar in general.

Response: We combined figures 1,2 and 3 according to the recommendation of respected reviewer. It remains to hope that it does not seem overloaded in this form

  1. Furthermore, evidence to support a role of lipid-binding and transfer proteins in long-distance transport of these phytohormones is scarce, please combine 5.1-5.3 and clarify that the involvement of LBTPs of long-distance transport of these plant hormones is speculative. 

Response: Sections 5.1-5.3 were combined

  1. The manuscript concerns the long-distance transport of plant hormones most under abiotic averments but not biotic environments, please clarify.

Response: There is almost no information on involvement of cytokinins and ABA in long distance signaling under biotic stress, while the case of jasmonates is the matter of debate. This is why we concentration on abiotic stresses. However responses to biotic stressed is mentioned: “It was shown that necrotrophic pathogen infection, insect herbivory and mechanical wounding induces accumulation of jasmonates at the site of injury as well as in distal, undamaged leaves”   

 .

My recommendations for revisions:

  1. The tittle of the manuscript is grammatically incorrect and can be changed to “The long-distance transport of some plant hormones and the possible involvement of lipid-binding and transfer proteins in this process”, given that Auxins, SAs and GAs were not considered by the manuscript.

Response: We accepted the title proposed by the respected reviewer. Only “in this process” was substituted with “in hormonal transport”, since another reviewer did not like “in this process”

  1. L 75, the subtitle should be “Long-distance signaling and transport of cytokinins ”.

Response: The proposed subtitle was accepted

  1. L251, in this section, jasmonates (JAs, jasmonate and related oxylipins) should be used and information on the shoot-root transport of JA precursor OPDA should be included.   

Response: In accordance with this competent recommendation, we mentioned “JAs, jasmonate and related oxylipins” in this section and added that “It has been shown that JA precursor OPDA (oxo-phytodienoic acid) and its derivatives, but not the biologically active form of JA-Ile, could be transferred from wounded shoots to healthy roots” with reference to the review of Wang et al.

  1. L109, the involvement of JA in SAR is questionable, the reference 109 do not support such a claim.

Response: We completely agree that involvement of JA in SAR is questionable. But still, such possibility is discussed. We are sorry for improper previous reference. We substituted it with Li M., Yu, G.; Ma, J.; Liu, P. Interactions of importers in long-distance transmission of wound-induced jasmonate. Plant Signal Behav. 2021, 16, 1886490. doi: 10.1080/15592324.2021.1886490. It supports such a possibility. To show that this is only a possibility, we added “can” to the sentence ( “jasmonates [109] … can participate in long-distance signaling during systemic acquired resistance”)

  1. L 440-443, the conclusion is not justified by the evidence. 

Response: We agree that this is only a suggestion. So the sentence is modified to make it less certain: “The possibility of interaction of jasmonates with LTPs is suggested by the results of their immunohistochemical localization with the corresponding antibodies on sections of pea roots (Fig. 2).”